# Intracellular neutralisation of rotavirus by VP6-specific IgG

**Sarah L. Caddy**[1,2]*, **Marina Vaysburd**[1], **Mark Wing**[1], **Stian Foss**[3,4], **Jan Terje Andersen**[3,4], **Kevin O'Connell**[1], **Keith Mayes**[1], **Katie Higginson**[1], **Miren Iturriza-Gómara**[5], **Ulrich Desselberger**[2], **Leo C. James**[1]*

**1** MRC Laboratory of Molecular Biology, Cambridge, United Kingdom, **2** Department of Medicine, University of Cambridge, Cambridge, United Kingdom, **3** Institute of Clinical Medicine and Department of Pharmacology, University of Oslo and Oslo University Hospital, Oslo, Norway, **4** Department of Immunology, University of Oslo and Oslo University Hospital, Oslo, Norway, **5** Centre for Global Vaccine Research, Institute of Infection and Global Health, University of Liverpool, Liverpool, United Kingdom

* slc50@cam.ac.uk; lcj@mrc-lmb.cam.ac.uk

**Data Availability Statement:** All relevant data are within the manuscript and its Supporting Information files.

**Funding:** This work was supported by the MRC (UK; U105181010), a Wellcome Trust Investigator

## Abstract

Rotavirus is a major cause of gastroenteritis in children, with infection typically inducing high levels of protective antibodies. Antibodies targeting the middle capsid protein VP6 are particularly abundant, and as VP6 is only exposed inside cells, neutralisation must be post-entry. However, while a system of poly immune globulin receptor (pIgR) transcytosis has been proposed for anti-VP6 IgAs, the mechanism by which VP6-specific IgG mediates protection remains less clear. We have developed an intracellular neutralisation assay to examine how antibodies neutralise rotavirus inside cells, enabling comparison between IgG and IgA isotypes. Unexpectedly we found that neutralisation by VP6-specific IgG was much more efficient than by VP6-specific IgA. This observation was highly dependent on the activity of the cytosolic antibody receptor TRIM21 and was confirmed using an *in vivo* model of murine rotavirus infection. Furthermore, mice deficient in only IgG and not other antibody isotypes had a serious deficit in intracellular antibody-mediated protection. The finding that VP6-specific IgG protect mice against rotavirus infection has important implications for rotavirus vaccination. Current assays determine protection in humans predominantly by measuring rotavirus-specific IgA titres. Measurements of VP6-specific IgG may add to existing mechanistic correlates of protection.

## Author summary

Rotavirus is the leading cause of gastroenteritis in children worldwide. Effective rotavirus vaccines have been available for over a decade, but detailed understanding of the immune response to rotavirus infection is essential for further improvement of vaccines. High levels of antibodies are made in response to infection, especially antibodies targeting the inner capsid protein VP6, but while both IgA and IgG isotypes are produced, previous work has focused predominantly on VP6-specific IgA. In this study we sought to evaluate the importance of VP6-specific IgG in rotavirus protection. As VP6-specific antibodies

Award to LCJ, a Wellcome Trust Clinical Research
Career Development Fellowship to SLC and a
Junior Research Fellowship from Magdalene
College, Cambridge to SLC. SF was supported by
the Research Council of Norway (grant no.
251037) while JTA was supported by the Research
Council of Norway (grant no. 287927), and the
South-Eastern Norway Regional Health Authority
(grant no. 2018052). The funders had no role in the
study design, data collection and analysis, decision
to publish, or preparation of the manuscript.

**Competing interests:** The authors have declared
that no competing interests exist.

target incomplete rotavirus particles inside cells, we developed a new assay to examine
how antibodies neutralise rotavirus intracellularly. We showed that neutralisation by
VP6-specific IgG was much more efficient than VP6-specific IgA, due to the activity of
the cytosolic antibody receptor TRIM21. This was confirmed using a mouse model of
rotavirus infection. Furthermore, mice with normal IgA levels but deficient in IgG had a
serious deficit in intracellular antibody-mediated protection. Our finding that VP6-spe-
cific IgG protect mice against rotavirus infection may be valuable for predicting whether
new rotavirus vaccines will work. Current assays to determine protection in humans focus
on measuring rotavirus-specific IgA titres. We propose that including measurements of
VP6-specific IgG may improve knowledge on correlates of protection.

## Introduction

Species A rotaviruses are a major cause of acute gastroenteritis in infants and young children
under 5 years of age worldwide [1]. Live attenuated rotavirus vaccines, licensed in 2006, have
been introduced with good efficacy and effectiveness in > 100 countries [2]. Antibodies are
rapidly generated in response to both natural infection and vaccination, and have been shown
to be essential for protection against future rotavirus-associated disease. However, the precise
nature of this protection is poorly understood, since rotavirus-specific antibodies of different
isotypes (IgG or IgA) target a diverse range of viral proteins [3].

Rotavirus is a triple-layered particle, and a large body of evidence indicates that antibodies
targeting the VP6 protein of the middle capsid layer play a key role in protection against rota-
virus infection. VP6-specific antibodies are produced to high titres in response to rotavirus
infection or vaccination [4,5], and mice experimentally infected with rotavirus are protected
by passive transfer of anti-VP6 antibodies [6,7] or anti-VP6 nanobodies [8]. Furthermore, a
number of VP6-based vaccine approaches show induction or enhancement of protective
immunity [9–13].

The mechanisms by which VP6 antibodies mediate protection have yet to be fully charac-
terised. VP6 is only exposed after the triple-layered viral particle has entered the cell and the
outer layer containing VP4 and VP7 has been uncoated to release the double-layered particle
(DLP) [14,15]. This strongly suggests that antibodies targeting VP6 act intracellularly. Previous
studies have shown that upon binding to DLPs, some anti-VP6 monoclonal antibodies inhibit
transcription [7,16–18]. However, although additional mechanisms of neutralisation inside
cells have been postulated [7], these have not been explored in detail. We have recently charac-
terised TRIM21 as a novel cytoplasmic antibody receptor [19,20], and therefore aimed to
investigate whether TRIM21 could be involved in antibody-mediated rotavirus neutralisation
inside cells.

Following natural infection in mice and humans, it is known that antibodies produced to
target VP6 are of both IgA and IgG isotypes [21–23]. To be functional, these antibodies must
have a means of co-localising with DLPs in the cytoplasm. For IgA isotypes, co-localisation has
been shown to occur following transcytosis of IgA in the gut epithelia [6,7], and a role for the
poly-immunoglobulin receptor (pIgR) has been demonstrated [18,24,25]. However, the mech-
anism of cellular entry for IgG from the circulation into epithelia is pinocytosis, not receptor-
mediated, and therefore is expected to occur at a lower rate [26]. Since any potential activity of
this isotype has generally been overlooked, we wanted to compare and contrast the activity of
both IgA and IgG VP6-specific antibodies *in vitro* and *in vivo*.

Due to the intracellular activity of VP6-specific antibodies, there is no high-throughput
functional assay to measure their activity. A simple assay to measure VP6-mediated

intracellular neutralisation *in vitro* would be a valuable tool to study interactions between VP6-specific antibodies of IgA and IgG subtypes with subviral particles.

In this study we present an *in vitro* assay that can efficiently measure intracellular neutralisation by VP6-specific antibody of any isotype. Using this assay, we show that VP6-specific antibodies utilize multiple mechanisms to block rotavirus replication. By extending this approach to *in vivo* infections in a mouse model we show that IgG plays an important role in protection.

## Materials and methods

### Cells and viruses

MA104 African green monkey fetal kidney cells (PHE culture collection) and immortalised murine embryonic fibroblasts (MEFs, described in [27]) from wild type and TRIM21 knock-out mice were maintained in DMEM supplemented with 10% fetal calf serum (FCS) and 100 I. U./ml penicillin and 100 mg/ml streptomycin. Following infection with rotavirus, cells were kept in serum-free medium, supplemented with trypsin (TPCK-treated, LS003740, Worthington Biochemical) at a final concentration of 1μg/ml.

Tissue culture-adapted simian rotavirus SA11 (G3P[1]) was used, propagated in MA104 cells as previously described [28]. Two murine rotavirus (G16P[3]) strains were used, propagated in neonatal mice; the EDIM strain was a kind gift from L. Svensson, and the EMcN strain was a kind gift from M. McNeal. All viral particles were activated with trypsin (as above) at 10μg/ml for 30 minutes at 37°C prior to *in vitro* infections.

### Antibody-mediated intracellular neutralisation assay

To rapidly introduce antibodies into cells, 2μl total volume of antibody or serum at a range of concentrations were electroporated using 2 pulses of 1400V, 20 pulse width, into $3 \times 10^5$ cells suspended in 13μl Neon® Resuspension buffer R using the Neon® Transfection System (Thermo Fisher Scientific). Cells were then resuspended in 330μl antibiotic-free media containing 10% FCS, before being added to the wells of a 96 well plate (in triplicate per test, 100μl per well). Cells were incubated at 37°C for 4 hours to become adherent to the plate and then washed once with PBS to remove any extracellular antibody. Next, 1000 fluorescent focus forming units (FFFU) virus in 100μl serum-free medium were added to each well. After 1 hour, an additional 50μl complete DMEM were added to each well, and infection was allowed to proceed for 16 hours at 37°C. For proteasome inhibition experiments, 1μM MG132 (Sigma) was added to wells 2 hours prior to infection, and maintained in media for 4 hours post-infection.

### Extracellular neutralisation assay

Serial dilutions of antibodies from 800ng starting amount in 12.5μl serum-free medium were incubated with 1000 FFFU trypsin-activated rotavirus in a 1:1 mixture for 1 hour at 37°C. The antibody-virus mixture was then diluted 1:10 in serum-free medium, and 100μl added in triplicate to MA104 cells seeded at $5 \times 10^4$ cells per well of a 96 well plate. After 1 hour, an additional 50μl complete DMEM was added to each well. Infection then proceeded for 16 hours at 37°C.

### Rotavirus quantification

Quantification of rotavirus was achieved using a fluorescent focus forming assay [28]. In brief, after a 16 hour infection as described above, medium was removed from plates, and plates were fixed with 1:1 methanol:acetone at -20°C for 20 minutes. Wells were then blocked with

100µl PBS—2%FCS for 20 minutes at room temperature. Primary antibody (Sheep polyclonal anti-rotavirus antibody, PA1-85845, Thermo Fisher Scientific) diluted 1:500 in PBS—2%FCS was added for 1 hour at room temperature. After 3 washes with PBS, 25µl Alexa-Fluor 488-conjugated anti-sheep IgG (Invitrogen) diluted 1:500 in PBS—2%FCS and compound Hoechst 33342 (20mM, Thermo Fisher Scientific) diluted 1:1000 were added to each well and plates were incubated at room temperature for 1 hour. Wells were washed three times with PBS, prior to imaging using a Nikon Eclipse Ti microscope and counting of fluorescent foci was performed using Nikon NIS-Elements version 4.4 software. For neutralisation experiments, counts of foci in wells with and without antibody were compared.

## Immunofluorescence

Monoclonal antibody (mouse) was electroporated into MEF cells as described above, then cells were plated on cover slips (Corning BioCoat, Poly-D-Lysine, 12 mm) in 24-well plates and allowed to adhere for 4 hours. Cells were infected with rotavirus at MOI 20 for 1 hour, then washed three times with PBS and fixed in 4% paraformaldehyde (PFA) (Thermo Fisher Scientific). Cells were permeabilized in 0.1% Triton X-100 in PBS (PBS-X) and blocked with 5% BSA (Thermo Fisher Scientific) in PBS-X. The electroporated antibody in cells was first stained with Alexa-Fluor 488-conjugated anti-mouse IgG (Invitrogen), then DLPs were stained with an anti-VP6 MAb conjugated to Alexa-Fluor 568. Antibody conjugation was performed according to manufacturer's instruction (Thermo Fisher Scientific). Hoechst 33342 diluted 1:1000 was added simultaneously with the final antibody incubation step. Both primary and secondary antibodies were diluted in 5% BSA-PBS-X, and incubated with cells for 1 hour at room temperature. After each antibody incubation, cells were washed three times with PBS, then once with water. The cover slips were carefully dried and adhered to slides using mounting medium (ProLong Diamond Antifade Mountant, ThermoFisher Scientific).

## Preparation and purification of rotavirus double-layered virus particles (DLPs)

The SA11 strain of rotavirus was inoculated onto cell culture monolayers of MA104 cells at an MOI of 0.1. DLPs were then purified according to previously published protocol [28]. In brief, when cell monolayers exhibited complete cytopathic effect, cultures were subjected to freeze-thawing three times. Virus was sedimented by ultracentrifugation in a Beckman ultracentrifuge at 100,000 x g for 1.5 hours at 4˚C. The resulting pellet was resuspended in TNC buffer (20mM Tris-HCl, 100mM NaCl and 1mM CaCl$_2$, pH8.0) and the cellular debris was removed by addition of Vertrel XF (Sigma) and vortexing. The suspension was then centrifuged at 4,100 x g for 30 minutes at 4˚C to separate virus from cell debris. Virus particles were sedimented by ultracentrifugation at 100,000 x g for 1.5 hours. Final purification of the pelleted virus was achieved by isopycnic CsCl gradient ultracentrifugation at 110,000 x g for 18 hours. Viral bands containing triple layered particles (TLPs) and DLPs were collected by needle puncture from the ultracentrifuge tube, and virus suspensions were dialysed against TNC buffer overnight at 4˚C. DLPs were quantitated using a NanoDrop spectrophotometer (Thermo Scientific), and purity was visualized by SDS-PAGE and Coomassie blue staining (Instant Blue, Expedeon).

## Generation of CRISPR knockout cell lines

Generation of knockout MA104 cell lines was achieved by electroporation of Cas9/gRNA ribonucleoprotein complexes (Cas9-RNP) along with single stranded guide RNAs. TracrRNA and crRNA against *TRIM21* were obtained from IDT. TracrRNA-crRNA complexes were assembled by incubating at 95˚C for 5 min followed by cooling on the benchtop to 20˚C. The RNA

complexes were mixed with recombinant Cas9 protein at a molar ratio of 1.2:1 and incubated at 37˚C for 10 min to form Cas9 RNP complexes. 50 pmol of Cas9-RNP complex was introduced into $8 \times 10^5$ MA104 cells using the Neon Transfection System (Invitrogen) with a setting of 2 pulses of 1400 V for 20 ms. 48 hr post electroporation, the cells were cloned by fluorescence-activated cell sorting into 96-well plates (1 cell/well). Knockout (KO) mutations were confirmed by western blotting for TRIM21 protein, and loss of function was confirmed by a 'Trim-Away'experiment as described in [29]; in brief anti-IKKα antibody (Abcam) was electroporated into WT and KO cells, then 3 hours later cells were lysed and probed by western blot for IKKα.

## Antibody production and purification

Hybridoma cell lines expressing monoclonal antibodies (MAb) recognizing rotavirus proteins were cultured in RPMI media supplemented with low IgG serum (Hyclone, GE Healthcare). The anti-VP6 IgA hybridoma 7D9 was a kind gift from H. Greenberg and J.E. Crowe. Hybridoma supernatant was collected and antibodies purified using an AKTA chromatography system (GE Healthcare). A protein G-agarose column was used for hybridomas producing VP4-specific IgG antibodies (1A9 and 7A12, first described in [30]) and a protein L-agarose column was used to purify VP6-specific IgA antibodies (7D9 and 2C5, first described in [6,30]). In brief, supernatant of hybridoma cultures was applied to the appropriate column at 4˚C, then following washing with 5 column volumes of PBS, antibody was eluted with 0.1M glycine buffer pH 2.0 and immediately pH adjusted with 1M Tris-HCl buffer pH 9.0. Coomassie blue staining of an SDS-PAGE gel confirmed purification of antibody, and following dialysis against PBS, the total MAb yield was determined using a NanoDrop spectrophotometer.

Polyclonal anti-DLP antibodies were raised in adult mice by subcutaneous immunization with 100μg purified DLPs. A boost immunization of 50μg DLPs was given subcutaneously 14 days later. Serum samples from immunized mice were collected 28 days after the initial immunization, and any infection challenge experiments were also performed at this time.

VP6-specific IgG was purified from human pooled IgG (Sanquin, The Netherlands) by affinity chromatography. DLPs were conjugated to agarose beads using AminoLink™ Plus Immobilisation Kit (ThermoFischer Scientific), then antibodies purified according to manufacturer's instructions.

Generation of mouse human-chimeric h7D9 MAbs was achieved by synthesizing cDNA fragments encoding 7D9 heavy chain variable regions (Genscript Inc, USA) that were subcloned in frame with the constant region of the human IgA1 or IgG1 heavy chain (HC) in the expression vectors pFUSEss-CHIg-hA1 and pFUSEss-CHIg-hG1 (Invivogen, USA). Similarly, a cDNA fragment encoding the 7D9 light chain (LC) variable region was synthesized and subcloned (Genscript) into pFUSEss-CLIg-hκ (Invivogen, USA). h7D9 IgA1 or IgG1 HC vectors were then transiently co-transfected with the LC vector into the Expi293 cell line using the Expi293 transfection kit (ThermoFisher, USA). The h7D9 IgG1 MAbs were purified using a CaptureSelect-$C_H1$ specific column (ThermoFisher) while the h7D9 IgA1 MAb was purified using a CaptureSelect-IgA specific column (ThermoFisher). Monomeric fractions were isolated by size exclusion chromatography using a Superdex 200 Increase 30/300 column and concentrated on AmiconUltra 100K spin columns (Millipore). Purified proteins were analysed by SDS-PAGE (ThermoFisher, USA). h9C12-IgG1-WT was produced as described [31].

## Rotavirus infection of mice

Neonatal BALB/c mice were infected with $5 \times 10^4$ FFFU of the Epizootic Diarrhoea of Infant Mice (EDIM) virus in 10μl total volume by oral gavage. Neonatal C57Bl/6 mice were infected

with 1.5 x $10^3$ FFU of murine rotavirus strain EMcN. Stool samples were collected by gentle palpation of pup abdomens once daily for 4 days. Samples were pooled, diluted 1:10 in PBS and clarified by centrifugation at 8000 x g for 5 minutes. Rotavirus infectivity was then quantified by fluorescent focus forming assay in MA104 cells and aliquots were stored at -80˚C until use.

TRIM21 knockout BALB/c mice were generated by back-crossing TRIM21KO C57Bl/6 mice [27] x wild type BALB/c mice for the first round, selecting for TRIM21 heterozygous and back-crossing for 7 more rounds to BALB/c selecting after each round TRIM21+/-. After 8 rounds, TRIM21 heterozygous mice were crossed to select pure TRIM21KO on Balb/c background. Adult wild type or TRIM21 knockout Balb/c mice were infected with 5 x $10^4$ FFU EDIM in 50µl total volume. Adult wild type or FcRn knockout C57Bl/6 mice (purchased from JAX) were infected with 15 FFFU EMcN in 50µl total volume. Faeces were collected once daily for the 7 days following infection and stored at -20˚C until further processing. Faecal pellets were diluted 1:10 in Earles Balanced Salt solution by weight, and clarified by centrifugation at 8000 x g for 5 minutes. Faecal levels of murine rotavirus virus were quantified using a commercial mouse rotavirus antigen ELISA kit (Cusabio) according to the manufacturer's instructions.

Serum samples were obtained via centrifugation of clotted blood samples collected either from saphenous venipuncture or cardiac puncture.

## Ethics statement

All mouse infection experiments were conducted in accordance with the 19.b.7 moderate severity limit protocol and Home Office Animals (Scientific Procedures) Act (1986). All animal work was licensed under the UK Animals (Scientific Procedures) Act, 1986 and approved by the Medical Research Council Animal Welfare and Ethical Review Body.

## ELISAs

Ninety-six-well polystyrene microtitre plates (Microlon ®, Greiner) were coated overnight at 4˚C with 100ng purified DLPs/well in carbonate/bicarbonate buffer. Plates were washed three times with 0.05% Tween 20 in phosphate buffered saline (PBS-T) before blocking with 5% skimmed milk-PBS-T for 1 hour at 37˚C, followed by three PBS-T washes. Plates were then incubated for 1 hour at 37˚C with dilutions of each serum sample in duplicate in 5% skimmed milk-PBS-T. After three washes with PBS-T, 50µl of horseradish peroxidase (HRP)-conjugated anti-mouse IgG antibody, or HRP-conjugated anti-mouse IgA (Bio-Rad) diluted 1:1000 in 5% skimmed milk PBS–T, were added to each well and incubated at 37˚C for 1 hour. The plates were washed four times with PBS-T, and bound antibody was detected by addition of 50µl tetramethylbenzidine (TMB, Invitrogen) followed by incubation at room temperature for 10 minutes. The reaction was stopped with 1M $H_2SO_4$ and the optical density (OD) was read at 450 nm with a microplate reader (PHERAstar). The background signal for each sample was determined by measuring the OD450 of serum samples incubated with carbonate/bicarbonate buffer alone.

For sandwich ELISAs, rotavirus polyclonal antibody (PA1-85845, Thermo Fisher Scientific) was coated onto plates at 20ug/µl in carbonate/bicarbonate buffer overnight at 4C. Plates were washed three times with PBS-T, then cell lysate from MA104 cells infected with SA11 or uninfected control were added to plates at 20ug/µl for 4 hours. Plates were then washed three times with PBS-T before blocking with 5% skimmed milk-PBS-T. The remainder of the protocol was as described above.

To study recombinant antibodies produced, ELISAs were performed with DLPs as described above, with the following modifications; bound recombinant antibodies were detected using either an alkaline phosphatase (AP) conjugated anti-human Fc-antibody from goat (#A9544), Sigma-Aldrich, USA) or AP-conjugated anti-human IgA F(ab)2 (#SAB3701226, Sigma Aldrich, USA). Binding was visualized by addition of phosphatase substrate (Sigma Aldrich) and the 405 nm absorption spectrum was measured using a Sunrise spectrophotometer (TECAN, Austria).

### Statistics

To compare differences between two groups of values, unpaired T-tests were performed. P values <0.05 were considered significant. Analyses were conducted using GraphPad Prism.

## Results

### Development of an intracellular neutralisation assay

To study the interaction between rotavirus VP6 protein and anti-VP6 antibodies intracellularly, an assay was developed to enable rapid introduction of antibody into the cytosolic compartment of cells. Serially diluted anti-VP6 mouse IgA monoclonal antibody (MAb) (800ng–50ng) or a non-specific control antibody (800ng) were electroporated into MA104 cells using the Neon Transfection System. Cells were then infected with rotavirus overnight, and the degree of virus neutralisation was measured by fluorescent focus forming reduction assay. Fig 1A and S1A Fig presents images from a typical neutralisation assay as captured by a Nikon Eclipse Ti microscope, and Fig 1B shows quantification of images from triplicate wells using Nikon NIS-elements analysis software. The relative level of infection was calculated based on the number of foci in the non-specific control well. Confocal microscopy confirmed that electroporation of anti-VP6 MAb resulted in rapid co-localisation with DLPs within 1 hour of rotavirus infection (Fig 1C, S1B Fig).

Comparison of two anti-VP6 IgA MAbs, 7D9 and 2C5, demonstrated that both were unable to neutralise rotavirus if incubated with virus extracellularly according to standard protocols, but inside cells both MAbs potently neutralised virus in a dose-dependent manner (Fig 1D). Conversely, MAbs directed against the outer capsid protein VP4 (1A9 and 7A12) were unable to neutralise rotavirus after electroporation into cells (Fig 1E), whereas previous studies had shown they can neutralize RV extracellularly [32].

### Characterisation of polyclonal antibodies mediating intracellular neutralisation

To determine whether it is the specific epitope specificity of 7D9 and 2C5 that allows intracellular neutralization or if polyclonal antibodies are similarly effective, we raised VP6 antisera by immunizing mice with purified DLPs (S2 Fig). Native sera from immunized mice or unimmunized controls were then electroporated into cells. Fig 2A demonstrates that effective intracellular neutralisation is mediated by anti-DLP antibodies. To test whether this neutralization is physiologically relevant and capable of providing protective immunity *in vivo* we performed infection and re-challenge experiments. Importantly, the data show that mice immunized with DLPs were protected from subsequent challenge with Epizootic Disease of Infant Mice (EDIM) virus (Fig 2B), with statistically significant differences in viral loads detected on days 3–6 post infection. This confirmed the importance of anti-VP6 antibodies *in vivo*.

We observed that the mice immunized with DLPs and protected from EDIM produced a significant IgG anti-VP6 response (Fig 2C). Conversely, no significant difference was observed

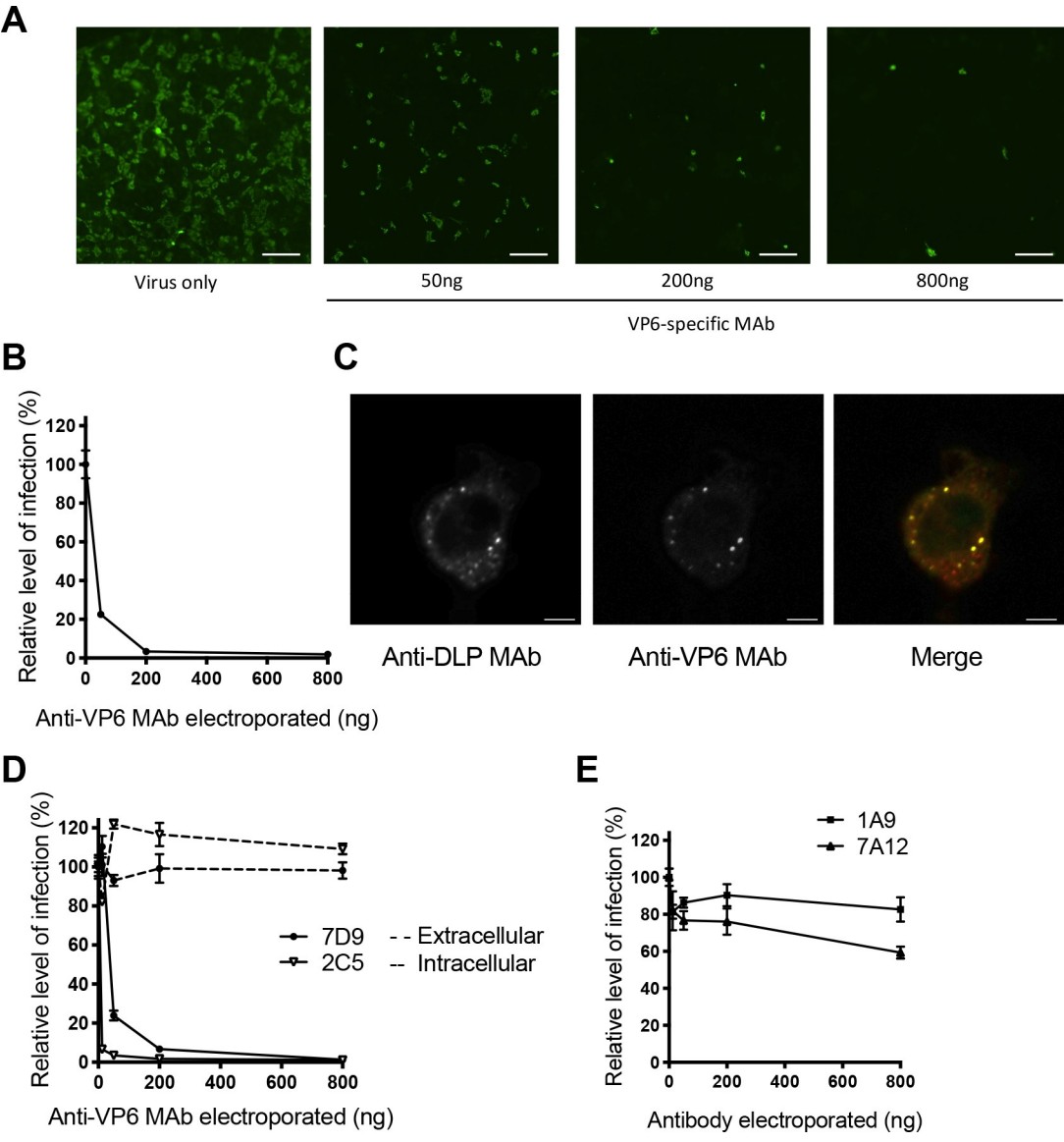

**Fig 1. Intracellular neutralisation assay.** (A) Images of rotavirus-infected cells following electroporation of anti-VP6 MAb. Rotavirus-infected cells were stained with sheep polyclonal anti-rotavirus antibody and subsequently Alexa-Fluor 488-conjugated anti-sheep Ig. Scale bar 200μm (B) Fluorescent foci in (A) captured on a Nikon Eclipse Ti microscope were quantified by NIS analysis software. (C) Confocal images of a rotavirus-infected murine embryonic fibroblast cell 1 hour post infection in the presence of electroporated anti-VP6 Mab, scale bar 10μm. (D) Comparison of extracellular versus intracellular neutralisation for two anti-VP6 MAbs, 7D9 and 2C5. (E) Testing for intracellular neutralisation mediated by MAbs recognising VP4 (7A12 and 1A9).

between IgA titres in naïve and immunized mice. It was expected that induction of IgG was due to the subcutaneous immunization route, but this raised the question of whether VP6-specific IgG or IgA were mediating the protection observed.

We reasoned that if VP6-specific IgG is important for protecting against rotavirus infection in natural infections, then it should neutralize rotavirus via the intracellular neutralization assay. DLPs were conjugated to agarose beads, and used to affinity purify VP6-specific antibody from IgG pooled from healthy human donors (Sanquin, The Netherlands). This demonstrated that VP6-specific IgG is produced in response to rotavirus infection in humans,

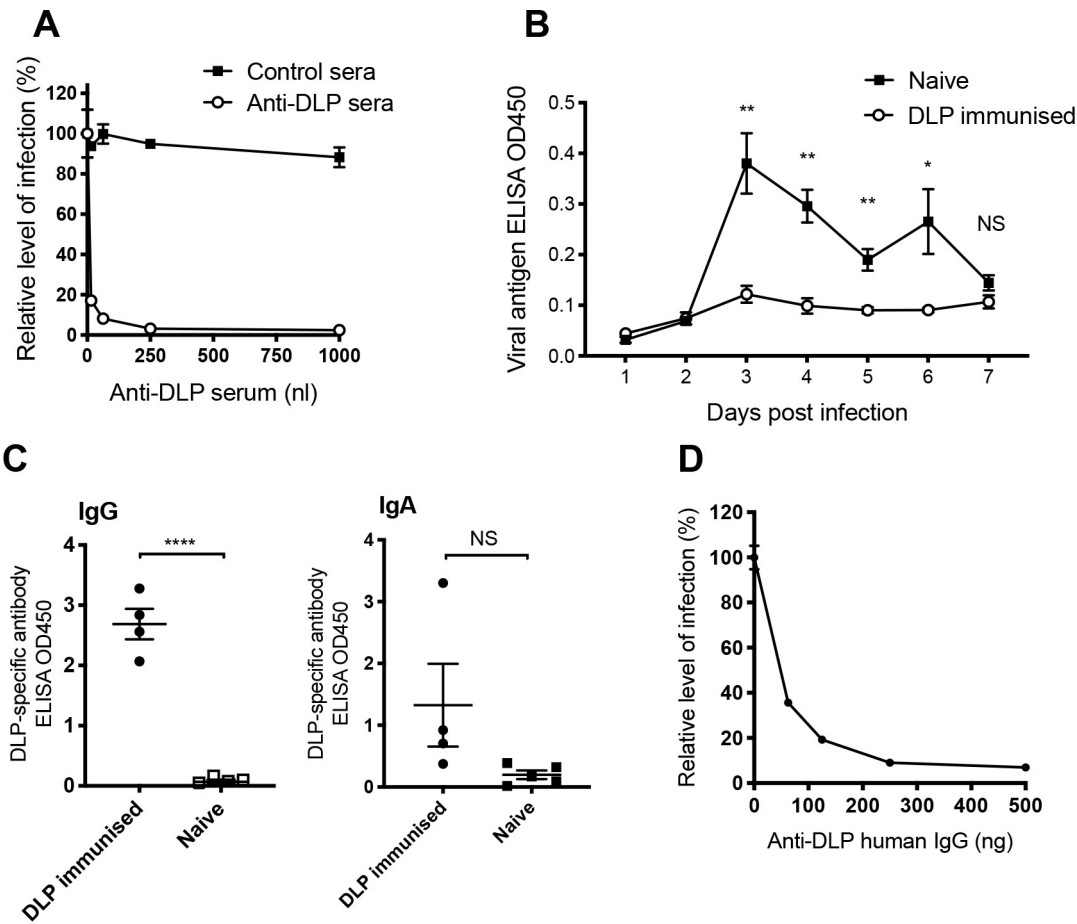

**Fig 2. Analysis of VP6-specific polyclonal immune responses in mice and humans** (A) Intracellular neutralisation of rotavirus infection by anti-DLP polyclonal serum versus control serum electroporated into cells. (B) Faecal antigen shedding as detected by ELISA on days 1–7 post infection in mice immunized with DLPs 2 and 4 weeks prior to infection, compared to naive mice (4–5 mice per group). (C) ELISAs to detect IgG and IgA polyclonal antibodies raised by DLP immunization in mice, each symbol representing one mouse. (D) Intracellular neutralisation of rotavirus (strain SA11) by VP6-specific IgG purified from human pooled IgG in MA104 cells. For all graphs, error bars represent standard error, **** p = < 0.0001, ** p = < 0.01, * p = < 0.05, NS not significant.

confirming previous reports that have identified VP6-specific IgG in naturally infected people [4,21,22]. Importantly, this VP6-purified human IgG efficiently neutralized rotavirus (strain SA11) in the intracellular neutralization assay conducted in MA104 cells (Fig 2D).

## Mechanism of intracellular neutralisation by IgG

To directly compare the relative potency of VP6-specific IgA and IgG, we produced recombinant class-switched 7D9 antibodies with either a human IgA1 or IgG1 heavy chain (hereafter referred to as hIgA and hIgG). Both monomeric antibodies were able to bind to DLPs in ELISA (Fig 3A and 3B), with significantly higher absorbance at every antibody concentration compared to non-specific mAb 9C12 as shown in Fig 3A. Previous studies using anti-VP6 IgA monoclonal antibodies have shown that blockade of mRNA egress from DLPs is a mechanism by which neutralisation can occur [7,16,18,33]. If the sole activity of 7D9 was pore blockade, it was theorized that the Fc region would have no effect on neutralisation within cells. To investigate this, mouse-human chimeric IgA 7D9 and IgG 7D9 were electroporated into MA104 cells

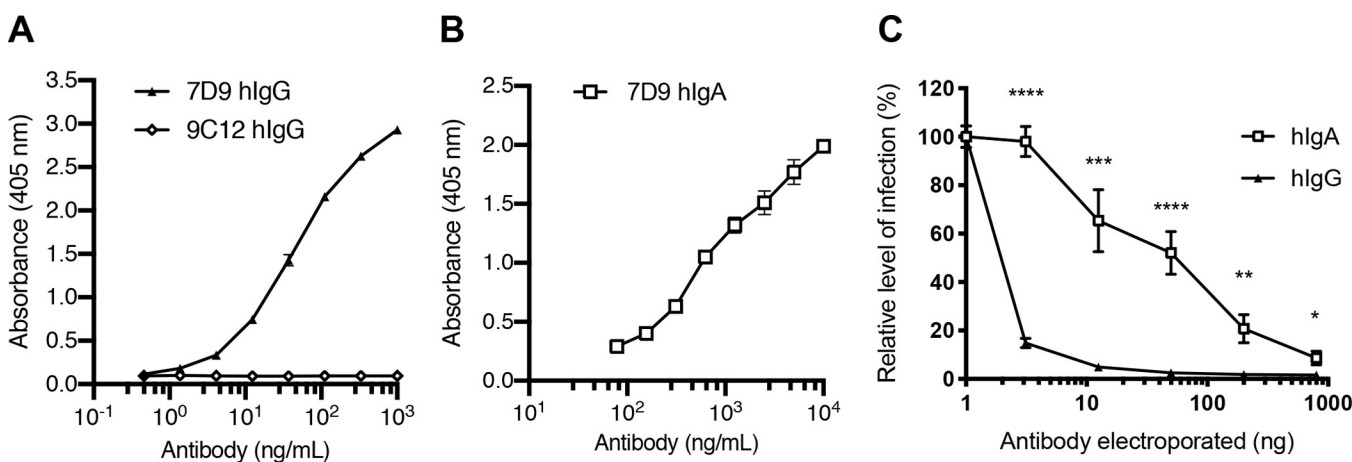

**Fig 3. Production and characterization of VP6-specific recombinant monoclonal antibodies.** (A) Mouse-human chimeric 7D9 hIgG binds to DLPs by ELISA while control antibody (9C12) does not. (B) Mouse-human chimeric 7D9 hIgA binds to DLPs by ELISA. (C) Intracellular neutralisation of rotavirus by VP6-specific 7D9 with human IgG1 Fc region compared to IgA1 region electroporated into cells. For all graphs, error bars represent standard error, **** $p = <$ 0.0001, *** $p = <$ 0.001, ** $p = <$ 0.01, * $p = <$ 0.05, NS not significant.

and intracellular neutralisation measured (Fig 3C). It was found that the IgG Fc region significantly enhanced neutralisation, with statistically significant differences identified at each concentration studied.

The intracellular Fc receptor TRIM21 is known to bind IgG with a substantially higher affinity than IgA [34]. Therefore, we hypothesised that the enhanced intracellular neutralisation mediated by VP6-specific IgG could be due to efficient recruitment of TRIM21. This Fc receptor is an E3 ligase that undergoes autoubiquitination when activated and targets the virus-antibody complex for degradation by the proteasome [19]. TRIM21 knockout MA104 cells were generated using CRISPR technology (S3 Fig), then used to evaluate intracellular neutralisation in the electroporation assay. There was no TRIM21-dependent component to intracellular neutralisation by 7D9 IgA (Fig 4A), but when using 7D9 IgG, a clear TRIM21-dependent phenotype was apparent (Fig 4B). Next, the Fc region of 7D9 IgG was engineered to prevent TRIM21 binding through the introduction of an H433A mutation previously shown to ablate TRIM21:IgG binding *in vitro* and their interaction in cells [31,35]. This mutation significantly decreased intracellular neutralisation mediated by IgG (Fig 4C). We also demonstrated a small TRIM21-dependent effect on intracellular neutralisation when low concentrations of sera from mice immunized with DLPs were electroporated into cells (S4 Fig). We showed that this neutralisation was partially diminished when cells were pretreated with 1μM of the proteasomal inhibitor MG132, supporting the hypothesis that proteasomal degradation is required for this neutralisation mechanism.

To extend these findings *in vivo*, wild type and TRIM21 knockout BALB/c mice were immunized with DLPs on days 0 and 14, then challenged with EDIM virus on day 28. Shedding of EDIM virus was then measured by faecal antigen ELISA. Viral shedding 3 days post infection is shown in Fig 4D; whilst immunization of wild type mice with DLPs significantly reduced peak viral shedding at day 3 ($p$ = 0.0007, unpaired t test), immunization of TRIM21 knockouts did not ($p$ = 0.211). Moreover, there was a significant difference when comparing across genotypes with immunized knockouts shedding more virus on day 3 than wild type equivalents ($p$ = 0.020). Virus shedding by naïve mice on day 3 post-infection was comparable for both wild type and TRIM21 knockout mice, confirming that TRIM21 antiviral activity is antibody-dependent. By 4 days post-infection the DLP-immunized TRIM21 knockout mice had controlled virus shedding to a similar degree to that of the wild type mice (Fig 4E).

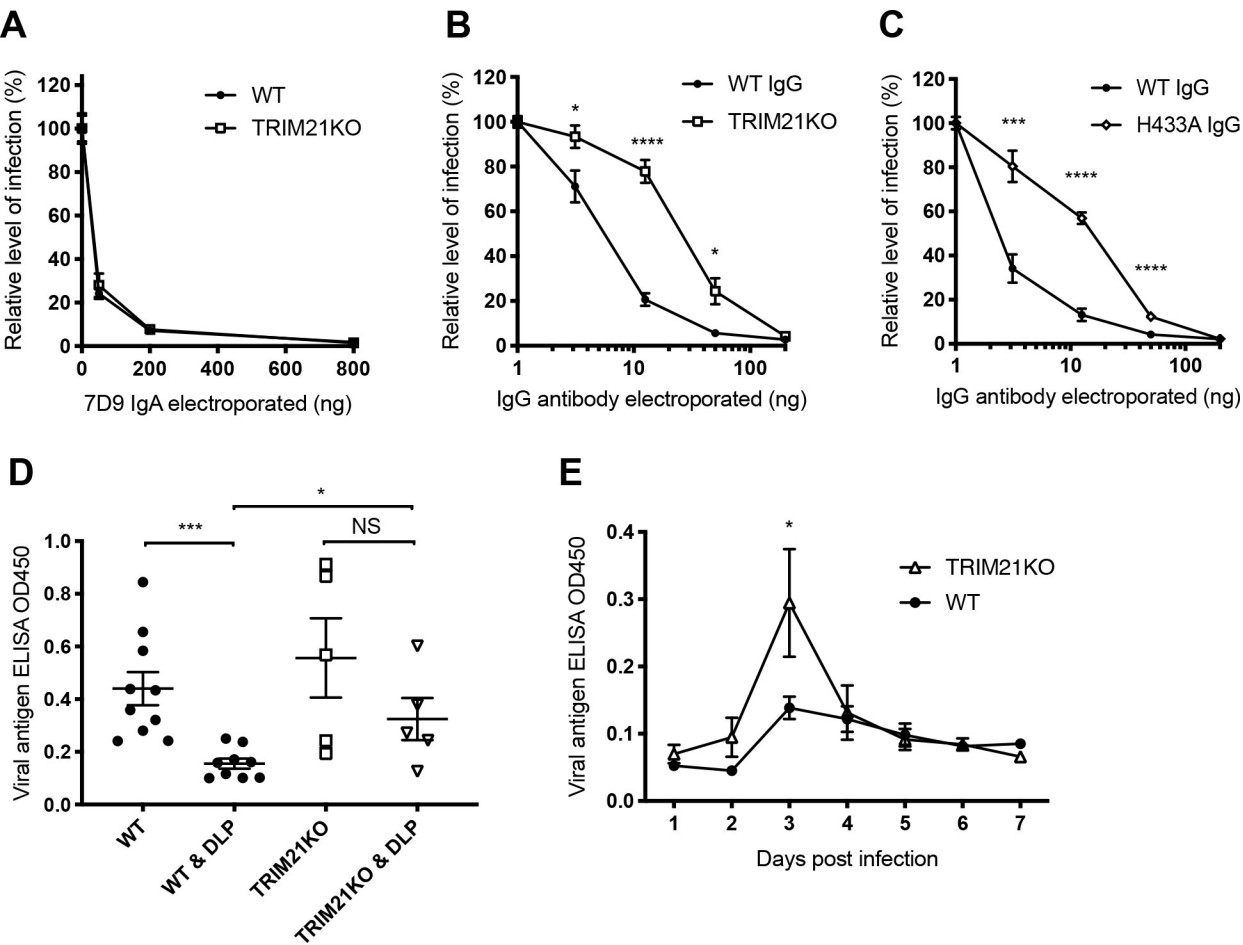

**Fig 4. Mechanisms of antibody-mediated intracellular neutralisation.** (A) Intracellular neutralisation of rotavirus by VP6-specific IgA in wild type (WT) and TRIM21 knockout (TRIM21KO) MA104 cells. (B) Intracellular neutralisation by VP6-specific IgG in WT and TRIM21KO knockout MA104 cells. (C) Intracellular neutralisation by WT or non-binding TRIM21 IgG (H433A). (D) Amounts of EDIM virus shedding three days post infection as detected by faecal antigen ELISA in naïve and pre-DLP-immunized, WT (9–10 mice per group) as well as TRIM21KO mice (5 mice per group). (E) EDIM shedding on days 1–7 post infection in WT and TRIM21KO BALB/c mice pre-immunized with DLPs. Horizontal lines represent mean and standard error, *** p = < 0.001, * p = < 0.05, NS not significant.

## VP6-specific IgG protects against rotavirus infection *in vivo*

While the predominance of IgG in the anti-VP6 response during DLP immunization was suggestive of its importance, we sought direct evidence of its physiological role by using a mouse model with reduced IgG but normal levels of other antibody isotypes. Mice deficient in the neonatal Fc receptor (FcRn) were selected for this purpose, as the absence of FcRn reduces antibody IgG levels by decreasing its half-life from 6–8 days to approximately 1 day [36]. In contrast to other antibody-deficient models, however, there is no change in antibody diversity or in B cell compartments [36,37].

WT and FcRn knockout mice (five mice per group) were immunized with DLPs on days 0 and 14, then challenged with EDIM virus on day 28. As the FcRn knockout mice used were on a C57Bl/6 background, we infected mice with an alternative murine rotavirus strain known to readily infect C57Bl/6 wild type mice (EMcN [38]). Fig 5A shows that following immunization, FcRn knockout mice generally had lower levels of serum IgG antibody than their wild type counterparts, although this difference was not statistically significant (unpaired t test *p* = 0.055). Importantly, when challenged with EMcN murine rotavirus, FcRn knockout mice

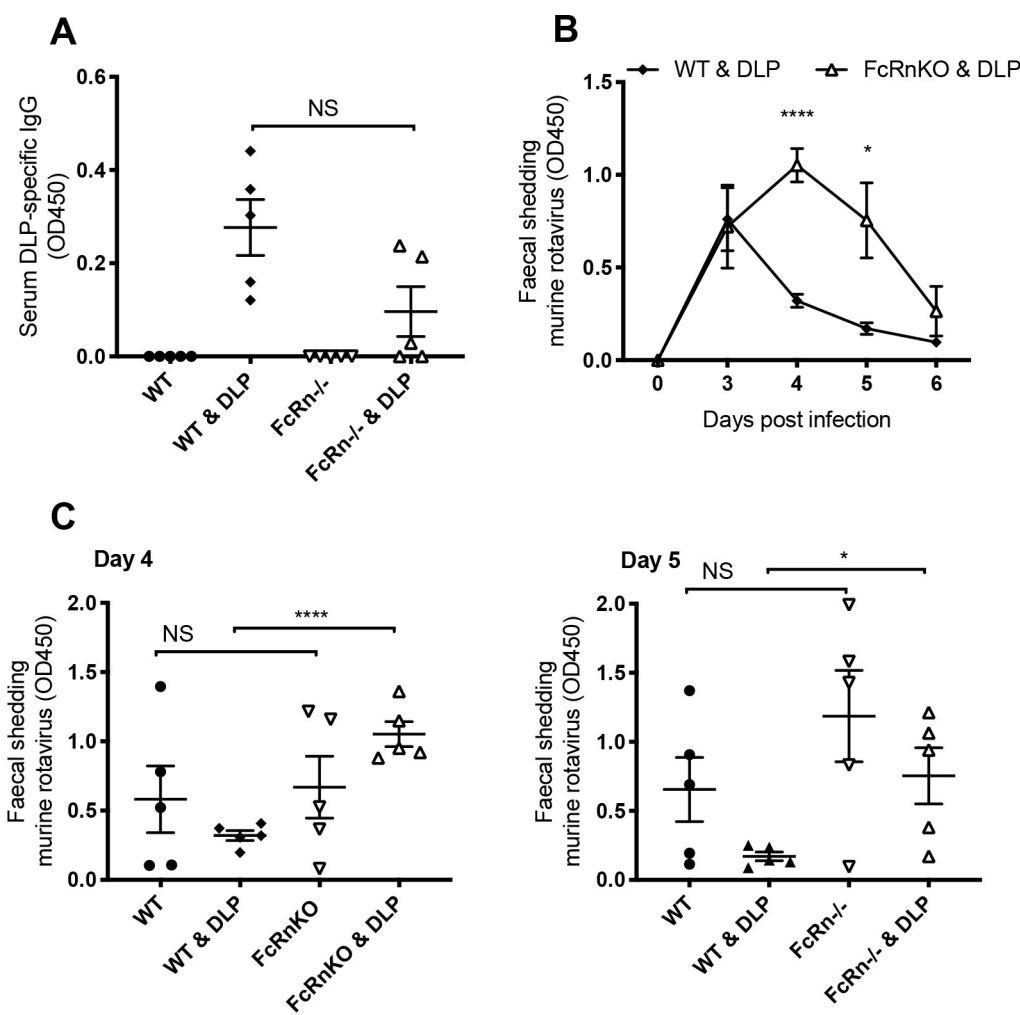

**Fig 5. Rotavirus infection in FcRn deficient mice.** (A) DLP-specific antibodies in serum of naïve and DLP-immunized, WT and FcRnKO mice (5 mice per group, each symbol representing one mouse). (B) Faecal antigen shedding on days 0–6 post infection as detected by ELISA in the mice pre-immunized with DLPs, comparing WT with FcRn knockout (FcRnKO) mice. (C) Amounts of EMcN virus shedding in faeces of the naïve and DLP-immunized, WT and FcRnKO mice, days 4 and 5 of experiment shown in panel (B). Horizontal lines represent mean and standard error, **** p = < 0.0001, * p = < 0.05, NS not significant.

were significantly less able to control virus shedding than wild type mice (Fig 5B). This was statistically significant on both days 4 and 5 post infection (unpaired t-test, $p < 0.0001$ and $p = 0.0217$ respectively), while infection levels in naïve mice of each genotype were comparable at these time points (Fig 5C).

## Discussion

Here we describe a new intracellular neutralization assay that allows direct comparison of neutralizing antibodies regardless of isotype. We have used the intracellular neutralization assay in combination with murine models of rotavirus infection to illustrate that VP6-specific IgGs have an important physiological role in neutralising rotavirus. The ability to study the interactions between antibodies and viruses like rotavirus inside cells has classically relied on transwell systems or lipid-based transfection reagents [7,25]. Whereas the former approach

presents a physiologically relevant system, it is relatively time-consuming to establish, and the efficiency of transfection systems is low. The electroporation-based intracellular neutralisation assay allows antibodies to be introduced directly into the cytoplasm rapidly and efficiently, supports high-throughput analyses and can be used with both MAbs and polyclonal antisera.

Whereas the focus of our study has been centered on rotaviruses, the electroporation methodology described can be used for other viruses that may be targeted by antibodies inside cells. For example, a number of MAbs specific for viral proteins only exposed intracellularly have been characterised for both influenza viruses and lymphocytic choriomeningitis virus [39–41]. These antibodies are non-neutralising *in vitro*, but *in vivo* are able to mediate protection from infection. Using the intracellular neutralisation assay to study the activity of these antibodies inside cells, and being able to easily measure their presence in clinical samples may be highly beneficial for future basic and translational research.

The intracellular neutralisation assay has demonstrated that the mechanisms by which intracellular antibodies are neutralising rotavirus are multifactorial. It has previously been shown that neutralisation can occur through blockade of VP6 pores of DLPs through which viral mRNAs egress, or through alteration of the pore structure when binding adjacently [7,16,18,33]. This phenomenon does not rely on any specific activity of the Fc region of antibodies, and indeed it is known than VP6-specific nanobodies can protect against rotavirus infection [8,42,43]. However, a recent study that examined the effect of fusing an IgG1 Fc domain fragment to a VP6 nanobody did show enhanced neutralisation [44], thus supporting the conclusion that multiple mechanisms are at play. We therefore sought to explore whether cellular factors could enhance this neutralisation, specifically the intracellular antibody receptor TRIM21. This was achieved using TRIM21 deficient cell lines, recombinant antibodies unable to bind TRIM21, and finally TRIM21 deficient mice. TRIM21 is an E3 ubiquitin ligase that is activated when it binds the Fc portion of cytosolic antibodies bound to virus, catalysing auto-ubiquitination. This ubiquitination targets the virus-antibody complex to the proteasome for degradation [19]. TRIM21 has been extensively characterised in the context of non-enveloped viruses with a single capsid layer, such as adenovirus and rhinovirus [45]. Non-entry blocking antibodies bound to the virus reach the cytoplasm via the viruses' natural entry pathway. However, we have now shown that TRIM21 can drive antibody-mediated neutralisation when the antibodies are most likely entering the cytoplasm independently of incoming virus. In summary, we propose that VP6-specific IgG are able to block viral replication via two different mechanisms; blockade of DLP pores as previously shown by numerous studies, and also TRIM21-mediated proteasomal degradation of subviral particles. It is predicted that TRIM21 activity is most valuable at low levels of VP6-specific antibody, as previous work has shown that TRIM21 can become activated by as few as two antibodies [20], whereas effective pore-blockade may require a higher antibody concentration. The involvement of TRIM21 in anti-VP6 mediated protection may explain why immunisation of some species with DLPs does not always provide protection. TRIM21 is an interferon stimulated gene (ISG) and its expression is induced by interferon, meaning that the presence of anti-VP6 antibodies may not be enough. To be protective there may also have to be an accompanying inflammatory response.

Multiple lines of evidence indicate that rotavirus-specific IgG is important for immune protection in several species. In a primate model, passive transfer of IgG protected against rotavirus infection [46], and in rabbits, IgG induced by vaccination with rotavirus-like-particles was protective against oral challenge with rotavirus [47]. In humans serum IgG levels correlate with protection in a number of studies [48], and furthermore, IgA-deficient individuals are still able to recover from rotavirus infection [49]. Rotavirus infection of FcRn deficient mice has also been performed, as a means of reducing total IgG levels *in vivo* [50]. FcRn functions as a recycling receptor, binding to IgG at low pH in endosomes and rescuing IgG from

degradation by the lysosomes. IgG titres in mice lacking FcRn are therefore approximately 30 times lower than in wild type mice [26]. While previous studies in FcRn deficient mice have revealed prolonged rotavirus shedding compared to wild type mice, these experiments were performed in naïve animals and the resulting primary antibody response will have contained entry neutralizing antibodies against the VP4 and VP7 proteins of the outer capsid [50]. To study the importance of VP6-specific IgG separately from all rotavirus-specific IgG, we tested FcRn-deficient mice that were immunized with DLPs. As IgA levels remain unchanged in FcRn deficient mice, the prolonged viral shedding we observed can be directly attributable to VP6-specific IgGs.

As VP6 is only exposed post-cell entry, our findings indicate that protective IgG must be entering epithelial cells independently of viral particles. IgA is known to be transported from the basolateral to the apical side of epithelial cells via pIgR-mediated transcytosis. IgA-containing endosomes fuse with virus-containing endosomes during transcytosis, allowing antibody-virus complexes to form. IgG can also be transported across epithelial cells by FcRn, following fluid-phase pinocytosis on the basolateral side [51,52]. We suggest that this is the means by which VP6-specific IgG can co-localise with a protein only exposed intracellularly. During trafficking by FcRn, analogously with pIgR trafficking, IgG-containing endosomes fuse with virus-containing endosomes. The antibodies bind DLPs within the endosome then accompany the virus into the cytosol.

While FcRn is known to be important for IgG homeostasis, we speculate that it may also enhance co-localisation of VP6-specific IgG with DLPs inside cells. This would be analogous to the proposed mechanism by which pIgR traffics anti-VP6 IgAs to virus-containing compartments during IgA-mediated neutralization [24,25]. FcRn localizes intracellularly in endosomes positive for Rab5 and EEA1 [53,54], which are the same intracellular compartments that rotavirus reaches upon entry into cells [55]. Following entry of IgG from the basolateral side of epithelia into pinocytic vesicles, IgG subsequently binds to FcRn in the endosomes as they acidify. All known strains of rotavirus reach maturing endosomes before uncoating their outer capsid [55], thus they would be expected to be present in the same cellular compartments as FcRn when binding to IgG occurs. We propose that FcRn could enhance the ability of VP6-specific IgG to meet DLPs, without which IgG is sorted into lysosomes. Further work on the potential role of FcRn supporting VP6-specific IgG binding to uncoated rotavirus particles in endosomes is warranted.

The finding that VP6-specific IgG can be protective in a mouse model of rotavirus infection raises questions regarding the use of current assays to establish correlates of protection for rotavirus vaccines. Moreover, our data highlight multiple mechanisms by which anti-VP6 antibodies can provide this protection. The most widely used correlate of protection for rotavirus in humans following vaccination is rotavirus-specific serum IgA titre [56,57]. This measure is often considered to be imperfect, with poor correlation between IgA seroconversion and protection from severe rotavirus-induced gastroenteritis in several studies [56,58]. We propose that omitting to measure rotavirus-specific IgG could be part of the explanation as to why serum IgA titres are not an optimal correlate of protection. Moving forwards, it is important that the role of VP6-specific IgG is evaluated for its ability to contribute to protection from rotavirus disease in the context of human infections and vaccination.

## Supporting information

**S1 Fig. Microscopy of electroporated cells.** A) Nuclei stained with Hoechst 33342 from the 4 different wells presented in Fig 1A, scale bar 200μm. B) Extended Fig 1C showing additional controls used for confocal images; only in the presence of both SA11 rotavirus and anti-VP6

Mab is co-localization observed. Scale bar 10μm.
(TIFF)

**S2 Fig. Purity of DLP preparation.** A) Image of viral bands after centrifugation at 110,000 x g for 18 hours on a CsCl gradient. B) Coomassie blue stained agarose gel of viral bands collected separately by needle puncture, demonstrating different protein composition of triple layered particles (TLP) and double layered particles (DLP). C) Western blot confirmation of the specificity of antibodies in sera from mice immunized with DLP in comparison with antibodies generated by mice infected with EDIM rotavirus; MA104 cells infected with rotavirus for 16 hours were lysed in SDS-PAGE loading buffer, the lysate separated by SDS-PAGE, then stained with Coomassie blue or western blotted with mouse sera.
(TIFF)

**S3 Fig. Confirmation of TRIM21 knockout of MA104 cells by CRISPR.** A) Western blot of whole cell lysate of one wild type (WT) MA104 clone and three TRIM21 knockout (KO) MA104 clones. B) Western blot showing ability of WT cells to degrade IKK when anti-IKK antibody is electroporated into cells ('Trim-Away'), whereas no degradation of IKK is mediated by TRIM21 KO cells (clone b).
(TIFF)

**S4 Fig. TRIM21-dependent intracellular neutralisation by DLP-specific polyclonal sera.** A) Intracellular neutralisation in wild type (WT) and TRIM21 knock out (TRIM21KO) cells by serially diluted anti-DLP polyclonal serum. B) Intracellular neutralisation of rotavirus by undiluted anti-DLP polyclonal serum and control serum in the presence of 1μM MG132. For both graphs, error bars represent standard error, $^*$ $p = < 0.05$.
(TIFF)

## Acknowledgments

We wish to thank Prof Harry Greenberg and Prof Franco Ruggeri for the kind gift of monoclonal antibodies, and Prof Lennart Svensson and Dr Monica McNeal for the gift of two strains of murine rotavirus. We also thank the Light Microscopy team and the Biological Services group at the MRC-LMB, as well as Olivia Stupart for technical assistance.

## Author Contributions

**Conceptualization:** Sarah L. Caddy, Ulrich Desselberger, Leo C. James.

**Formal analysis:** Sarah L. Caddy, Ulrich Desselberger, Leo C. James.

**Funding acquisition:** Sarah L. Caddy, Stian Foss, Miren Iturriza-Gómara, Leo C. James.

**Investigation:** Sarah L. Caddy, Marina Vaysburd, Mark Wing, Stian Foss, Kevin O'Connell, Keith Mayes, Katie Higginson, Miren Iturriza-Gómara.

**Methodology:** Sarah L. Caddy, Stian Foss, Jan Terje Andersen, Ulrich Desselberger, Leo C. James.

**Project administration:** Leo C. James.

**Resources:** Jan Terje Andersen, Ulrich Desselberger, Leo C. James.

**Supervision:** Jan Terje Andersen, Miren Iturriza-Gómara, Ulrich Desselberger, Leo C. James.

**Writing – original draft:** Sarah L. Caddy, Ulrich Desselberger, Leo C. James.

**Writing – review & editing:** Sarah L. Caddy, Ulrich Desselberger, Leo C. James.

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
