## [Decision Letter · Decision Letter 0]

19 Mar 2020

Dear Sarah

Thank you very much for submitting your manuscript "Intracellular neutralisation of rotavirus by VP6-specific IgG" for consideration at PLOS Pathogens. As with all papers reviewed by the journal, your manuscript was reviewed by members of the editorial board and by several independent reviewers. In light of the reviews (below this email), we would like to invite the resubmission of a significantly-revised version that takes into account the reviewers' comments.

We cannot make any decision about publication until we have seen the revised manuscript and your response to the reviewers' comments. Your revised manuscript is also likely to be sent to reviewers for further evaluation.

Sincerely,

Gregory Tiao, M.D.

Associate Editor

PLOS Pathogens

Ana Fernandez-Sesma

Section Editor

PLOS Pathogens

Kasturi Haldar

Editor-in-Chief

PLOS Pathogens

orcid.org/0000-0001-5065-158X

Michael Malim

Editor-in-Chief

PLOS Pathogens

orcid.org/0000-0002-7699-2064

The authors have addressed an important topic regarding intracellular neutralization of rotavirus however the manuscript is not acceptable at the current stage since the reviewers had some serious concerns/questions. If the authors can address the reviewers queries we might reconsider.

Reviewer's Responses to Questions

**Part I - Summary**

Reviewer #1: The manuscript by Caddy SL et al. examined the role of anti-rotavirus (RV) non-surface structural protein VP6 antibodies in neutralizing viruses intracellularly. The authors found that anti-VP6 IgG, when electroporated into MA104 cells, blocked RV infection. Based on the authors’ previous publication (Clift D, Cell, 2018), they went on to examine the function of TRIM21 in this process in vitro and in vivo. The paper focuses on an interesting and under-studied aspect of mucosal immunity and has relevance to the study of RV correlates of protection in humans. The manuscript, in general, is well written. However, key controls are missing for a number of reagents used in the paper, undermining its credibility. The connection between in vitro cell culture (Figs. 1-3) and in vivo knockout mice (Figs. 4-5) is also weak. My specific comments are as follows:

Reviewer #2: The study “Intracellular neutralization of rotavirus by VP6-specific IgG” by Caddy and colleagues investigates the role of intracellular neutralization of RV VP6 by IgG class antibodies. There are several novel aspects in this study including the development of an intracellular neutralization assay to compare IgA and IgG isotype mediated effects on RV infection. The main finding here is that IgG isotype antibodies directed to DLP/VP6 can protect against RV infection. Overall, these findings are important in the field and present a novel insight into RV immunity. As presented it is difficult to evaluate how significant the effects are and whether the data supports the main conclusions.

Reviewer #3: The authors developed an intracellular neutralisation assay to examine how antibodies neutralise rotavirus in the cytoplasm. They report that neutralisation by VP6-specific IgG was more efficient than by VP6-specific IgA and neutralization was dependent on the activity of the cytosolic antibody receptor TRIM21.

**Part II – Major Issues: Key Experiments Required for Acceptance**

Reviewer #1: Major points:

1) In Fig. 1A, is there a reason why “fluorescent focus forming reduction assay” (Results, Line 350) was used as opposed to traditional plaque forming unit assays? One potential caveat of this imaging-based assay is that the anti-VP6 antibodies may compete with the detection antibody, giving rise to false positive results. Also, viral antigens were stained at 16 hours post infection (Materials and Methods, Line 155). The authors are encouraged to perform plaque assays within a single replication cycle (8 hours) to determine the neutralization capacity of these anti-VP6 IgGs.

2) It is stated in Materials and Methods, Line 210, that CsCl gradient ultracentrifugation was used to purify double-layered particles (DLPs), which were later used for immunization studies (Fig. 2). Did the authors test infectivity of their DLP preparation? It is possible that a trace amount of triple-layered particle contaminants induced anti-VP4 and VP7 antibodies. Have the authors tested their anti-DLP sera by western blot or ELISA to rule out immunity to either VP4 or VP7?

3) Data in Fig. 3 is not compelling and can be contributed by antibody binding affinities. For instance, at 103 ng/ml, 7D9 hIgG reached an OD of 3.0 (Fig. 4A) whereas 7D9 hIgA was only at 1.2 (Fig. 4B). In theory, the reduced blocked of infectivity by hIgA could be due to a weaker binding rather than the Fc region (Fig. 4C). To definitely show that the Fc of IgG is required, the authors should generate a Fab fragment and demonstrate the loss of activity.

4) The authors should elaborate or at least speculate on the mechanism of intracellular IgG-mediated neutralization. Is it transcriptional inhibition by blocking the mRNA portals on DLPs or is it DLP degradation?

5) It is hard to imagine that if anti-VP6 IgG is important for protection in humans, there will be no epidemiological data to support that. In the vaccinated children cohort, do anti-VP6 IgG levels correlate with protection?

Reviewer #2: 1. The authors have presented data on effects of Ig on RV throughout the paper as a percentage effect (relative level of infection) on a 10-log scale. This is very confusing and hampers evaluation of claims on how potent the neutralization effects are likely to be. This reviewer suggests presenting the viral titers on a 10-log scale – or as a less clear alternative to show the relative percentages with errors on a linear scale.

2. The statistical significance should be marked for every data panel – including in cases where there is no significant difference. The results and discussion should be in accordance with the statistical interpretation.

3. Figure legends are missing details to allow the reader to understand what was done in each experiment. This should be included.

Reviewer #3: Figure 1A. Electroporation is knows to be toxic to cells. Was LDH or another method used to measure cell viability? Nuclei should be stained to show that the cell density is similar in each of the wells. The text indicates that a non-specific control antibody was also electroporated in this experiment. Was the amount of virus quantified in the control antibody electroporated cells? I’m assuming this would be indicated in Figure 1B. What is the titer of virus at 100% infection? It is critical to determine the cytotoxicity of electroporation in all the electroporation experiments.

Figure 1C. There can be a slow exchange of secondary antibody in these types of experiments since 2 different mouse antibodies were used for this staining. Were proper controls performed to show that the detection is specific?

In many of the figures error bars are not indicated. How many times were experiments repeated and how many technical replicates were performed within each experiment?

Figure 2. It is very difficult to purify DLP that do not generate antibodies against the outer capsid proteins. How pure were the immunizing DLPs? Did the mice generate antibodies against VP4 or VP7? Was this tested by western blot or another method to determine whether the mice generated antibodies against VP4 or VP7?

Figure 2B. It has been shown that mice immunized with non-replicating virus-like particles composed of VP2 and VP6 are protected from challenge. The author’s result with DLPs is expected. However, other animal models such as rabbits, pigs or cows are not protected from disease or shedding following immunization with VP2/VP6 VLPs. It is only in mice that DLPs or VP2/VP6 VLPs is protective and this should be discussed.

Figure 2C. The amount of IgG vs IgA cannot be accurately determined by this ELISA. The amount of each antibody type has to be determined based on a standard curve.

Figure 2D. What virus and cells were used for this experiment? This was not described in the figure legend or the text. Was a western blot used to verify that antibody against VP6 was the only antibody isolated with the DLPs conjugated to agarose beads?

How was the TRIM21 knockout confirmed in the MA104 cells?

Although significance is stated in the text for Figs. 4B and C, statistics are not shown on the figure. If TRIM21 is the mechanism by which 7D9 IgG is mediating the neutralization, why are higher concentrations of the antibody able to neutralize either in the absence of TRIM21 or when the interacting amino acid is mutated? Was a confirmatory experiment performed to show that introduction of the H433A mutation in 7D9 IgG abrogated the antibody interaction with TRIM21?

For the experiments shown in Figs. 4D and E, were RV-specific antibody titers determined as well as the immunoglobulin type of the antibody?

The figure legend for 4E is unclear. Does this represent EDIM shedding days 1-7 post infection? As it reads, the mice were immunized with DLPs on days 1-7 post infection. Are any of the time points significantly different?

In all the mouse experiments, how many mice were in each group? This should be stated in the figure legend although I’m assuming that the number of markers shown in the graph indicates a separate mouse. For figure 5A, it appears there are only 3 mice in the FcRn & DLP group.

In the discussion, the authors indicate that TRIM21 is activated when it binds the Fc portion of cytosolic antibodies bound to virus, catalysing auto-ubiquitination. This ubiquitination targets the virus-antibody complex to the proteasome for degradation. If this is the case, can the authors show that the electroporated VP6 mAb is degraded in MA104 cells and this is proteasome-dependent by the addition of MG132? In addition, the mAb should not be degraded in the TRIM21 KO MA104 cells. Is proteasomal degradation the fate of the DLPs in the electroporated cells? Is this the mechanism of “neutralization”?

**Part III – Minor Issues: Editorial and Data Presentation Modifications**

Reviewer #1: Minor points:

1) Abstract, Line 28, VP6 is the “middle” capsid protein, not the “inner” capsid protein.

2) Abstract, Line 35, there was no data to support that the neutralization is “in the cytoplasm” or endosome/membrane-associated.

3) Introduction, Line 73, “Species A” should be “Group A”.

4) Materials and Methods, Lines 157 and 165, the subtitle should be in bold to be consistent with the other parts.

5) Materials and Methods, Line 182, “electroporated into MEF cells”. However, I cannot find the use of MEFs in the paper.

6) Scale bars should be provided for Figs. 1A and 1C.

7) In Fig. 1D, the label for 7D9 extracellular should be circles instead of squares.

8) The labels for Figs. 4A and B are very confusing. The x-axis should be made consistent with Ig classes: IgA for 4A and IgG for 4B.

9) In Fig. 4A, how were the TRIM21 KO MA104 cells validated? By western blot or Sanger sequencing? Please specify.

10) Figs. 4D-E, do the authors have evidence that the effect of TRIM21 is T cell-independent?

11) In Fig. 5A, what are the levels of DLP-specific IgG in the intestinal tissues?

12) There was no data to support that IgG actually get inside cells. To bridge the MA104 results with the FcRn KO mice data, can the authors generate an MA104-FcRn stable cell line, which would be more physiologically relevant than electroporation.

13) Figure legends, Lines 823 and 825, (B) and (C) were arranged in the wrong order.

14) Figure legends, Line 828, there was no panel (D) in the Fig. 5.

15) Statistics should be provided for all figure panels.

Reviewer #2: See comments on figure legends and log axes above.

Reviewer #3: (No Response)

PLOS authors have the option to publish the peer review history of their article (what does this mean?). If published, this will include your full peer review and any attached files.

Reviewer #1: No

Reviewer #2: No

Reviewer #3: No
---

## [Decision Letter · Decision Letter 1]

5 Jun 2020

Dear Sarah L. Caddy,

Thank you very much for submitting your manuscript "Intracellular neutralisation of rotavirus by VP6-specific IgG" for consideration at PLOS Pathogens. As with all papers reviewed by the journal, your manuscript was reviewed by members of the editorial board and by several independent reviewers. The reviewers appreciated the attention to an important topic. Based on the reviews, we are likely to accept this manuscript for publication, providing that you modify the manuscript according to the review recommendations.

Sincerely,

Gregory Tiao, M.D.

Associate Editor

PLOS Pathogens

Ana Fernandez-Sesma

Section Editor

PLOS Pathogens

Kasturi Haldar

Editor-in-Chief

PLOS Pathogens

orcid.org/0000-0001-5065-158X

Michael Malim

Editor-in-Chief

PLOS Pathogens

orcid.org/0000-0002-7699-2064

Reviewer Comments (if any, and for reference):

Reviewer's Responses to Questions

**Part I - Summary**

Reviewer #1: This is a revision of the original manuscript by Caddy SL et al. Specifically, the authors made extensive edits to the text and added new supplementary figures 1-4 to address the potential concerns. Most of the points raised were satisfactorily addressed. However, this reviewer finds the response to Question 1 insufficient.

**Part II – Major Issues: Key Experiments Required for Acceptance**

Reviewer #1: Plaque assays unable to “provide more rapid results (16 hours v 3-4 days)” and “a highthroughput method using the high content Nikon Eclipse Ti microscope.” seem irrelevant here. Other alternative approaches such as focus forming unit assays and quantitative PCR are also readily available to confirm the authors’ findings. The fact that “sheep polyclonal anti-rotavirus antibody that targets all viral proteins as our detection reagent” does not give support to the validity of the assay used in Fig. 1.

**Part III – Minor Issues: Editorial and Data Presentation Modifications**

Reviewer #1: Two minor points: 1) “dependant” should be “dependent” in marked document, lines 425 and 905; 2) statistics should be applied to Figs. 3A, 3C, 4A-C, and S4A-B.

PLOS authors have the option to publish the peer review history of their article (what does this mean?). If published, this will include your full peer review and any attached files.

Reviewer #1: No
---

## [Editor Report · Decision Letter 2]

22 Jun 2020

Dear Dr Caddy,

We are pleased to inform you that your manuscript 'Intracellular neutralisation of rotavirus by VP6-specific IgG' has been provisionally accepted for publication in PLOS Pathogens.

Best regards,

Gregory Tiao, M.D.

Associate Editor

PLOS Pathogens

Ana Fernandez-Sesma

Section Editor

PLOS Pathogens

Kasturi Haldar

Editor-in-Chief

PLOS Pathogens

orcid.org/0000-0001-5065-158X

Michael Malim

Editor-in-Chief

PLOS Pathogens

orcid.org/0000-0002-7699-2064
---

## [Editor Report · Acceptance letter]

28 Jul 2020

Dear Dr Caddy,

We are delighted to inform you that your manuscript, "Intracellular neutralisation of rotavirus by VP6-specific IgG," has been formally accepted for publication in PLOS Pathogens.

Best regards,

Kasturi Haldar

Editor-in-Chief

PLOS Pathogens

orcid.org/0000-0001-5065-158X

Michael Malim

Editor-in-Chief

PLOS Pathogens

orcid.org/0000-0002-7699-2064